# Hydrochemical Fluxes in Bulk Precipitation, Throughfall, and Stemflow in a Mixed Evergreen and Deciduous Broadleaved Forest

**Lei Su** [1,2] **, Changming Zhao** [2] **, Wenting Xu** [2] **and Zongqiang Xie** [2,*]

[1] International Joint Research Laboratory for Global Change Ecology, School of Life Sciences, Henan University, Kaifeng 475004, China; sulei123456a@126.com

[2] State Key Laboratory of Vegetation and Environmental Change, Institute of Botany, Chinese Academy of Sciences, Beijing 100093, China; zhaochangming@ibcas.ac.cn (C.Z.); xuwt@ibcas.ac.cn (W.X.)

[*] Correspondence: xie@ibcas.ac.cn; Tel.: +86-10-6283-6284

**Abstract:** Rainfall is one of the primary sources of chemical inputs in forest ecosystems, and the basis of forest nutrient cycling. Mixed evergreen and deciduous broadleaved forests are currently one of the most threatened ecosystems due to their sensitivity to anthropogenic climate change. As such, understanding the hydrochemical fluxes of these systems is critical for managing their dynamics in the future. We investigate the chemistry of bulk precipitation, stemflow and throughfall in a mixed evergreen and deciduous broadleaved forest in the Shennongjia region of Central China. Mean nutrient concentrations in throughfall and stemflow were higher than in bulk precipitation. Stemflow ion fluxes from deciduous tree species were greater than those for evergreen tree species because of the differences in bark morphology and branch architecture. Throughfall and stemflow chemistry fluctuated dramatically over the growing season. Nitrate nitrogen and ammonium nitrogen were retained, while other elements and compounds were washed off or leached via throughfall and stemflow pathways. Our findings will facilitate a greater understanding of nutrient balance in canopy water fluxes.

**Keywords:** throughfall; stemflow; water chemistry; macronutrients input; Hubei Province

## 1. Introduction

Nutrient cycling can directly determine the productivity and stability of forest ecosystem, and is the basis for maintaining forest ecosystem stability and sustainability [1]. Nutrient cycling is tightly linked to hydro-eco-pedological processes. Research on nutrient dynamics would contribute to the assessment of forest hydrology, and exploring various nutrients with the distribution of precipitation is critical for understanding ecosystem services and functioning [2–4]. With the aggravation of climate change, shifts in precipitation regime are becoming more frequent and severe, the projected changes in precipitation amount and seasonality can profoundly alter forest hydrological processes [5]. The alteration in precipitation regime will likely influence the nutrient cycling of forest ecosystems. Therefore, understanding the relationship between precipitation and nutrient cycling is highly important.

Atmospheric precipitation is an important source of nutrients in forest ecosystems. In nutrient-limited areas, precipitation can be the sole source of nutrient inputs, and can influence the growth and succession of forest communities [6–8]. After reaching the forest canopy, precipitation washes off particles and gases deposited on the vegetation surface during dry periods and fog events (dry and occult atmospheric deposition), mobilizes plant secretions, changes the vegetation canopy's original chemical form, and acts as a medium for the transfer of nutrients to soil. Some elements or compounds are easily leached from plant tissues (e.g., base cations) and precipitation is therefore

enriched with these substances during its passage through the canopy, while other substances are taken up in return (e.g., protons, ammonium) [9,10].

Bulk precipitation can be partitioned into interception loss, throughfall, and stemflow as it passes through the forest canopy. Interception loss is defined as the fraction of rainfall intercepted by the canopy and then evaporates back into the air. The latter two fractions both reach the ground surface as understory rainfall [11,12]. Throughfall is the major component of understory rainfall, making this component a direct nutrient source for forest plants and microorganisms. Throughfall is also a key regulator of the biogeochemical cycle of the Earth's surface [13–15]. As reported by Hansen [16], solute concentrations of throughfall are closely linked to distance from the nearest tree stem, with closer distances being associated with higher solute concentration.

While stemflow volume is typically much less than throughfall, it is still an important point-scale water flux [17,18]. Stemflow can directly change the physicochemical properties of the root zone and accelerate the redistribution of nutrients in forest ecosystems. Hence, stemflow is recognized as a key factor regulating hydrochemical characteristics [19–21]. Rainfall partitioning involves complex interactions among multiple vegetation factors [22–24]. Due to variation in branching structure, bark morphology, and canopy epiphytes, plants vary greatly in terms of absorption, assimilation, and leaching abilities of nutrients; stemflow nutrient fluxes vary greatly among different tree species [25–27]. In addition to the value in determining the availability of water for tree survival and growth, stemflow can also affect physical and biological components of forest ecosystem, which can lead to further differentiation in the soil microclimate [18,28,29]. Variation in stemflow chemistry thus poses a major challenge in the study of forest ecosystem nutrient budgets. Specifically, coniferous species retain more inorganic nitrogen than deciduous species do, but have a greater tendency to leach metals, such as potassium ion ($K^+$), calcium ion ($Ca^{2+}$), and magnesium ion ($Mg^{2+}$) [9]. Variation in stemflow yield for evergreen and deciduous broadleaved tree species has been well documented [30]. However, differences in stemflow chemistry between evergreen and deciduous broadleaved tree species are still not well understood.

Mixed evergreen and deciduous broadleaved forests are a common transitional forest type between deciduous broadleaved forests and evergreen broadleaved forests, and are one of the most vulnerable ecosystems to climate change due to their small geographic ranges [31]. Hydrochemical characteristics of rainfall will greatly determine the survival and future development of these forests. With increased research focusing on these forests, there is a pressing need to better understand the chemistry of the hydrological fluxes of the forest. Within this context, the aim of this work was to test the hypothesis that the chemistry of rainfall fractions are greatly altered after passing through the canopy; the functional group (i.e., evergreen vs. deciduous) can substantially change stemflow nutrient fluxes. To accomplish this, we conducted a field study focusing on the hydrochemical fluxes of bulk precipitation and throughfall. Meanwhile, the stemflow chemistry between the main evergreen and deciduous tree species was also studied. Our specific objectives were: (1) to compare the changes in chemical properties of rainfall after passing through the canopy of a mixed evergreen and deciduous broadleaved forest; and (2) to identify the differences in stemflow chemical composition between evergreen and deciduous tree species in this transitional forest.

## 2. Materials and Methods

### 2.1. Site Descriptions

This study was conducted during the growing season (May to October) of 2014 within a mixed evergreen and deciduous broadleaved forest in Central China. The study area was located in the Shennongjia Biodiversity Research Station of the Chinese Academy of Sciences, in the northwestern portion of Hubei Province in Central China (110°28′27″ E, 31°18′23″ N). The mean elevation of the research site was 1650 m above sea level (Figure 1). Importantly, the research sitewas located within the transition zone between the northern subtropical zone and the mid-subtropical zone. The area has

been noted as an important global biodiversity hotspot [31]. The climate is humid with a hot summer. According to Köppen Classification [31], it is classified as warm temperate climate. Average annual precipitation, mean temperature and relative humidity are 1350 mm, 10.6 °C, and 75%, respectively. The majority (85%) of the precipitation is distributed from May to October.

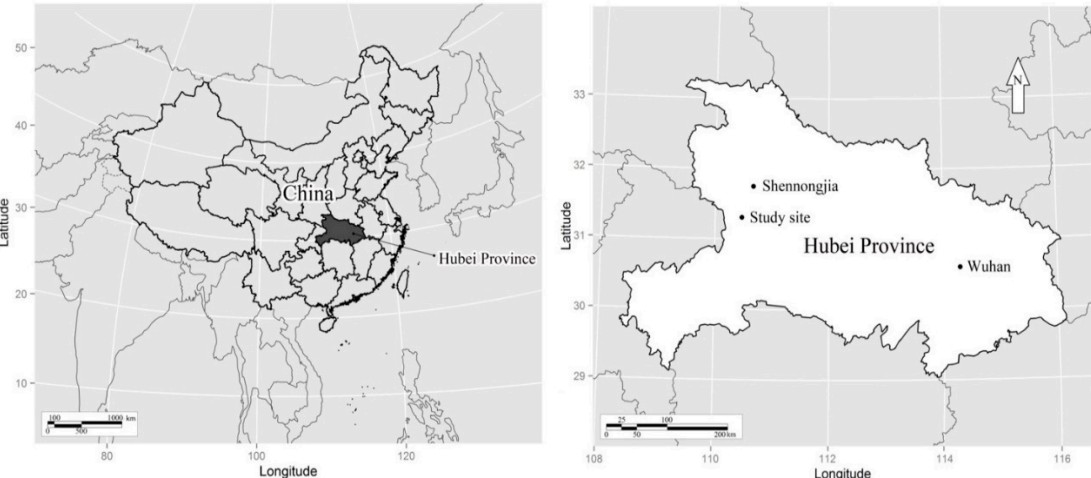

**Figure 1.** Location of the study site.

The study site is highly representative of mixed evergreen and deciduous broadleaved forests in China, especially in terms of the high biodiversity of the area. The density and total basal area of trees with height >2 m was 2407 ha$^{-1}$ and 42.56 m$^2$ ha$^{-1}$, respectively. Canopy cover during the growing season is, on average, approximately 85%. The dominant species included *Cyclobalanopsis multinervis* WC Cheng & T. Hong, *Cyclobalanopsis oxyodon* (Miq.) Oerst., *Fagus engleriana* Seem. and *Quercus serrata* var. *brevipetiolata* Bl. Herbs were sparsely distributed under the tree layer, with percent coverage less than 20%. The main species of understory herbs included *Fargesia spathacea* Franch., *Indocalamus tessellates* (Munro) Keng f., *Carex tristachya* Thunb., *Epimedium brevicornu* Maxim., *Viola verecunda* A. Gray, *Tiarella polyphylla* D. Don.

### 2.2. Study Design

Data were collected within a 40 m × 40 m plot within the mixed evergreen and deciduous broadleaved forest. Bulk precipitation was monitored in an open area adjacent to the study plot, where three gutter collectors were located. The width, length, and depth of the collectors were 0.2 m, 1.0 m, and 0.2 m, respectively. An automatic weather station (Model MAWS301, HydroMet$^{TM}$ system, Vaisala Corp., Helsinki, Finland) was located 500 m away from the experimental site, and was employed to measure rainfall amount and intensity. To measure throughfall, we placed 36 additional gutter collectors across the plot. Throughfall collectors were located 50 cm above the forest floor and were evenly distributed at fixed positions throughout the growing period. Measurements unit was mm, equivalent of 1 liter of the water per square meter.

Trees were categorized into five diameter at breast height (DBH) size classes (i.e., 6–10 cm, 10–14 cm, 14–18 cm, 18–22 cm and >22 cm). The species we examined were two evergreen species (i.e., *C. multinervis* and *C. oxyodon*) and two deciduous species (i.e., *F. engleriana* and *Q. serrata*). For each of the four studied species, we chose five trees that we considered to be representative of the five DBH classes (20 trees in total). Stemflow was collected using plastic tubing of 25 mm internal diameter. The top part of the tube was first cut from the middle, and was then stapled onto the trunk with small nails and sealed with silicon sealant. The bottom part of the plastic tube was kept intact, and the collected stemflow was funneled into a plastic jug on the forest floor. To prevent throughfall and litterfall from entering stemflow collectors, a plastic shelter was tied at the joint of the tube and the jug (Figure 2).

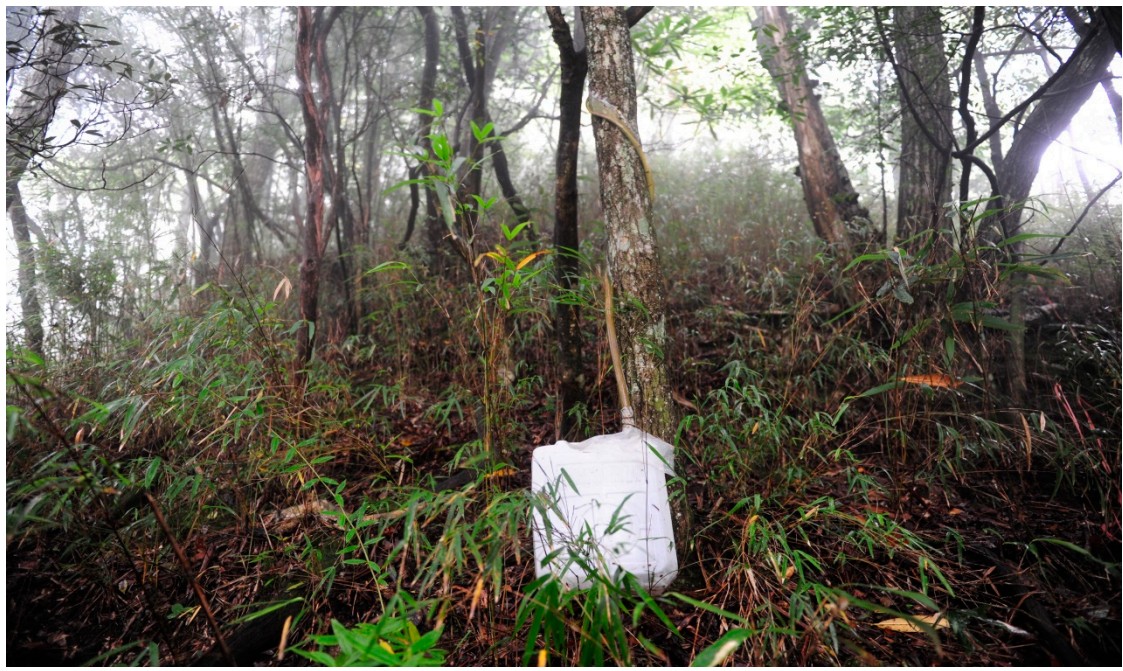

**Figure 2.** Stemflow collector with connected precipitation tank.

We checked and cleaned the precipitation tanks and stemflow collectors prior to each rainfall event. If any signs of leakage were detected, appropriate maintenance measures were conducted immediately. An individual rainfall event was defined as a measurable rainfall amount separated from the adjacent rainfall input by the time required for canopy to dry. All samples were then measured and collected either 2 h after the cessation of a rain event or the next morning for events that extended into or occurred during nighttime hours.

*2.3. Sampling Design*

We collected samples from three rainfall events every month. After the cessation of each rainfall event, all of the three bulk precipitation samples and the 20 stemflow samples were collected immediately. Ten throughfall samples were randomly chosen from the 36 throughfall collectors and were used for chemical analysis. In total, there were 594 individual water samples which were analyzed in the laboratory.

The electrical conductivity (EC) and pH of the water samples were determined immediately using a portable water quality detector (model YSI 6600V2, YSI Inc., Yellow Springs, OH, USA). Before laboratory analysis, water samples were filtered through cellulose acetate filter (pore size 0.45 μm, Ahlstrom ReliaDisc$^{TM}$, Whatman Corp., Maidstone, UK) into plastic bottles, and preserved in a laboratory refrigerator (−20 °C). The concentrations of chloride ion ($Cl^-$), sulfate radical ($SO_4^{2-}$), and nitrate radical ($NO_3^-$) were then determined using ion chromatography (Ion chromatograph, model Thermo IC-1500, Thermo Corp., Waltham, MA, USA). The concentration of ammonium ion ($NH_4^+$) was determined using the colorimetric method (Automated Discrete Analyzers, model SmartChem140, Alliance Corp., France), and the concentrations of $K^+$, sodium ion ($Na^+$), $Ca^{2+}$, $Mg^{2+}$ were determined using nitric acid acidification-ICP spectrometry (Inductively Coupled Plasma Spectrometer, model Thermo iCAP 6300, Thermo Corp., Waltham, MA, USA).

*2.4. Data Analysis*

The average value of the 36 throughfall collectors was considered as a measure of stand throughfall depth. The stand stemflow depth (mm) was computed following the equation used by Fan et al. [32]:

$$SF = \sum_{i=1}^{n} \frac{m \cdot S_i}{A \cdot 10^4} \qquad (1)$$

where *SF* represents the estimated stand stemflow depth (mm), *m* represents the number of trees belonging to a certain DBH class, $S_i$ represents the averaged stemflow volume (mL) of sampled trees in a certain DBH class, *A* represents the area of the study site (m²), and *n* represents the number of DBH classes (*n* = 5 in the current study).

The volume-weighted mean per event *E* ($VWM_E$) of bulk precipitation, throughfall, and stemflow were calculated as follows:

$$VWM_E = \frac{\sum\limits_{n=1}^{i} C_{i,E} \cdot V_{i,E}}{\sum\limits_{n=1}^{i} V_{i,E}} \qquad (2)$$

where $VWM_E$ represents the mean concentration of a chemical element or compound (mg·L$^{-1}$), $C_{i,E}$ represents the concentration at collector *i* for event *E* (mg·L$^{-1}$), and $V_{i,E}$ represents the rainfall, throughfall, and stemflow amount at collector *i* for event *E* (mm) [33,34].

The input of a given nutrient was estimated according to Filoso et al. [35]:

$$D = \frac{C \cdot V}{100} \qquad (3)$$

where *D* represents the input of a certain element or compound (kg ha$^{-1}$), *C* represents the mean concentration of a certain element or compound (mg·L$^{-1}$), and *V* represents the total volume of bulk precipitation, throughfall or stemflow (mm).

The total input of certain element or compound is the sum of the solute in the throughfall and stemflow, and the difference between the value of total input and bulk precipitation input is regarded as the rainfall loading of certain element or compound [6,36].

The stemflow nutrient concentrations among different tree species were analyzed with one-way analysis of variance (ANOVA). The sample amounts of bulk precipitation, stemflow and throughfall were inconsistent, so the nonparametric Kruskal–Wallis test was used to compare the nutrient concentrations among these rainfall fractions. All statistical analyses were performed using SPSS 18.0 (SPSS Inc., Chicago, IL, USA).

## 3. Results

*3.1. Hydrological Fluxes*

Between 1 May and 31 October, 48 discrete rainfall events resulted in a total of 1374.0 mm of rainfall. The precipitation during this period was 20% higher than mean growing-season precipitation. The maximum and minimum rainfall inputs were recorded in August (461.8 mm) and May (109.3 mm), respectively. Throughfall accounted for the largest portion of bulk precipitation volume across all six months of monitoring (the yield of throughfall was 1335.1 mm during the monitoring period), and the proportion occurring as throughfall ranged from 72.1% to 88.7% with a mean of 84.8%. Interception loss was the second rainfall fraction, and total interception loss was 225.5 mm, representing 14.3% of bulk precipitation. Stemflow yield was very small, the amount was only 13.4 mm, accounting for 0.5% of bulk precipitation in May, and 1.0% in August, which accounted for a total of 0.9% of bulk precipitation across the study period (Figure 3).

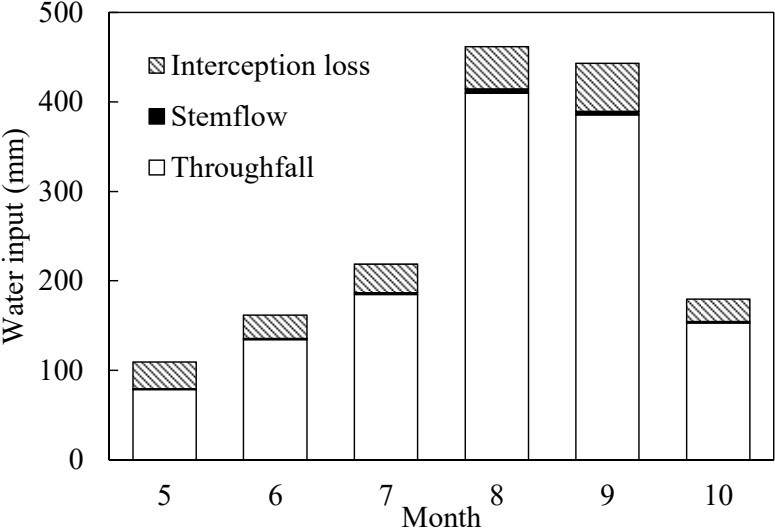

**Figure 3.** Monthly input of water via throughfall, stemflow and interception loss.

### 3.2. Chemistry of Canopy Water Fluxes

The mean EC of bulk precipitation was 4.7 $\mu$S cm$^{-1}$, while the EC of throughfall and stemflow were 11.7 $\mu$S cm$^{-1}$ and 14.6 $\mu$S cm$^{-1}$, respectively. These values were 2.48 and 3.10 times greater than that for bulk precipitation, respectively. The higher EC observed for throughfall and stemflow can be ascribed to changes in elements and compounds concentrations. The pH of bulk precipitation ranged from 7.37 to 7.52, and increased to 7.43–7.62 and 7.54–7.83 for throughfall and stemflow. There was no significant difference among these three rain fractions ($p > 0.05$). Most of the elements and compounds were more concentrated in throughfall and stemflow, except for $NO_3^-$ and $NH_4^+$, which were more concentrated in bulk precipitation. There was no significant difference in $NO_3^-$ concentrations among the three water fluxes ($p > 0.05$). $NH_4^+$ concentration in bulk precipitation was close to that for throughfall ($p > 0.05$), and both of these were lower than stemflow ($p < 0.05$).

Throughfall increased $Cl^-$ (222%), $SO_4^{2-}$ (179%), $Ca^{2+}$ (105%), $K^+$ (477%), $Mg^{2+}$ (256%), and $Na^+$ (50%) in comparison with bulk precipitation. The concentrations of these solutes was greater in stemflow than in throughfall. Compared to throughfall, $Cl^-$, $SO_4^{2-}$, $Ca^{2+}$, $K^+$, $Mg^{2+}$, and $Na^+$ were enhanced in stemflow by 25%, 18%, 56%, 80%, 31%, and 107%, respectively (Table 1).

**Table 1.** Chemical characteristic of water in bulk precipitation, throughfall and stemflow during the monitoring period.

| Chemical Variable | Concentrations (mg L$^{-1}$) | | |
|:---:|:---:|:---:|:---:|
| | **Bulk Precipitation** | **Throughfall** | **Stemflow** |
| EC * | 4.7 ± 0.1 [a] | 11.7 ± 0.4 [b] | 14.6 ± 0.6 [c] |
| pH | 7.37–7.52 [a] | 7.43–7.62 [a] | 7.54–7.83 [a] |
| $Cl^-$ | 0.27 ± 0.02 [a] | 0.64 ± 0.04 [b] | 0.80 ± 0.07 [c] |
| $NO_3^-$ | 0.43 ± 0.08 [a] | 0.35 ± 0.06 [a] | 0.57 ± 0.09 [a] |
| $SO_4^{2-}$ | 0.98 ± 0.05 [a] | 2.73 ± 0.21 [b] | 3.21 ± 0.26 [c] |
| $NH_4^+$ | 0.17 ± 0.01 [a] | 0.19 ± 0.01 [a] | 0.32 ± 0.03 [b] |
| $Ca^{2+}$ | 0.75 ± 0.08 [a] | 1.54 ± 0.11 [b] | 2.41 ± 0.15 [c] |
| $K^+$ | 0.43 ± 0.05 [a] | 2.48 ± 0.35 [b] | 4.46 ± 0.26 [c] |
| $Mg^{2+}$ | 0.09 ± 0.01 [a] | 0.32 ± 0.02 [b] | 0.42 ± 0.02 [c] |
| $Na^+$ | 0.10 ± 0.01 [a] | 0.15 ± 0.02 [b] | 0.31 ± 0.05 [c] |

* the unit of electrical conductivity (EC) is $\mu$S cm$^{-1}$. Different letters in the same row indicate statistical differences at $p < 0.05$.

The EC values of stemflow of the two deciduous species (i.e., *F. engleriana* and *Q. serrata*) were significantly higher than those for the two evergreen species (i.e., *C. multinervis* and *C. oxyodon*) ($p < 0.05$). This can be explained by the subsequent differences in solute concentrations between the different life history strategies. Specifically, all elements and compounds other than $Mg^{2+}$ were more concentrated in the stemflow of deciduous species than in evergreen species (Table 2). The pH values of the two deciduous were slightly but insignificantly higher than those of the two evergreen species ($p > 0.05$).

**Table 2.** Chemical characteristic of water in stemflow of the four tree species.

| Chemical Variable | Evergreen | | Deciduous | |
|---|---|---|---|---|
| | *Cyclobalanopsis multinervis* | *Cyclobalanopsis oxyodon* | *Fagus engleriana* | *Quercus serrata* |
| EC * | 14.3 ± 0.1 [a] | 14.3 ± 0.1 [a] | 14.9 ± 0.1 [b] | 16.3 ± 0.2 [c] |
| pH | 7.50–7.64 [a] | 7.57–7.71 [a] | 7.57–7.80 [a] | 7.73–7.95 [a] |
| $Cl^-$ | 0.78 ± 0.02 [a] | 0.75 ± 0.02 [a] | 0.80 ± 0.03 [a] | 0.94 ± 0.05 [b] |
| $NO_3^-$ | 0.38 ± 0.02 [a] | 0.60 ± 0.07 [ab] | 0.62 ± 0.07 [ab] | 0.74 ± 0.09 [b] |
| $SO_4^{2-}$ | 3.18 ± 0.11 [b] | 2.93 ± 0.08 [a] | 3.27 ± 0.07 [b] | 3.98 ± 0.14 [c] |
| $NH_4^+$ | 0.27 ± 0.01 [a] | 0.26 ± 0.01 [a] | 0.37 ± 0.02 [b] | 0.49 ± 0.03 [c] |
| $Ca^{2+}$ | 2.25 ± 0.13 [a] | 2.19 ± 0.11 [a] | 2.54 ± 0.15 [ab] | 3.02 ± 0.17 [b] |
| $K^+$ | 4.16 ± 0.23 [a] | 4.07 ± 0.17 [a] | 5.21 ± 0.22 [b] | 5.03 ± 0.20 [b] |
| $Mg^{2+}$ | 0.51 ± 0.02 [a] | 0.37 ± 0.03 [b] | 0.33 ± 0.04 [b] | 0.38 ± 0.02 [b] |
| $Na^+$ | 0.24 ± 0.03 [a] | 0.26 ± 0.01 [a] | 0.37 ± 0.04 [b] | 0.47 ± 0.03 [c] |

* the unit of EC is µS cm$^{-1}$. Different letters in the same row indicate statistical differences at $p < 0.05$.

### 3.3. Temporal Dynamics of Water Chemistry

Electrical conductivity (EC) of bulk precipitation was lower than throughfall and stemflow, and the EC in stemflow of the two deciduous tree species was higher than that for the two evergreen tree species (Figure 4). Concentrations of elements and compounds in throughfall and stemflow were more variable over time than concentrations in bulk precipitation. Concentrations of most elements and compounds ($Cl^-$, $NO_3^-$, $NH_4^+$, $Ca^{2+}$, $K^+$, $Mg^{2+}$ and $Na^+$) were relatively low at the beginning of the vegetation season and became higher over time. Element and compound concentrations dropped dramatically during the middle of the growing season and recovered towards the end of the growing season. Most of the elements and compounds in throughfall and stemflow were lowest in August, but were relatively higher by the end of the growing season.

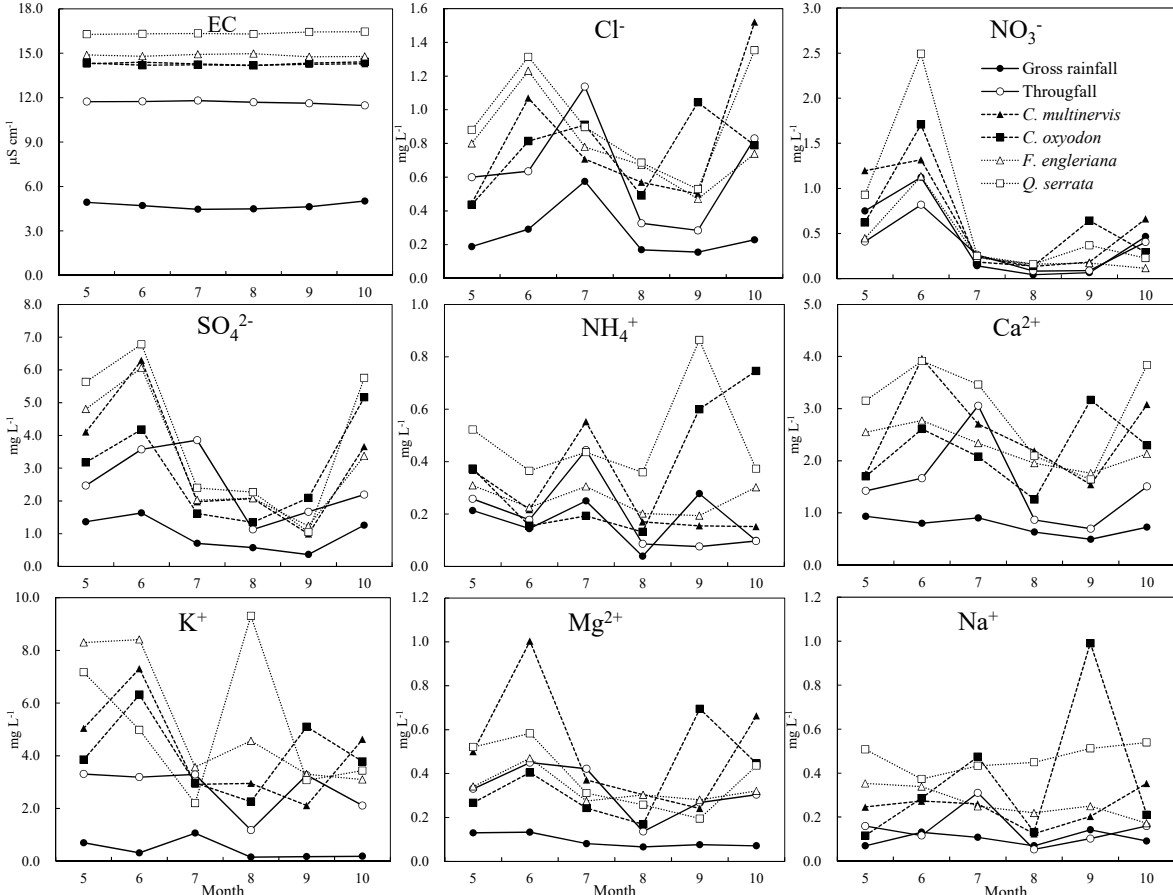

**Figure 4.** Monthly variation in chemical characteristic in waters of different type pathways and species of trees.

### 3.4. Nutrient Input

We observed large variation in nutrient input within the mixed evergreen and deciduous broadleaved forest. The highest nutrient input via bulk precipitation was $SO_4^{2-}$, while $K^+$ ranked as the largest input for throughfall and stemflow (Table 3). The total input of nutrient fluxes during the study period suggested the following trend: $K^+ > SO_4^{2-} > Ca^{2+} > Cl^- > Mg^{2+} > NO_3^- > NH_4^+ > Na^+$. Rainfall loading of nutrient fluxes indicated slight differences in total input, with the amount of rainfall loading being ranked as follows: $K^+ > SO_4^{2-} > Ca^{2+} > Cl^- > Mg^{2+} > Na^+ > NO_3^- > NH_4^+$. The rainfall loadings of the first six elements and compounds were positive, suggesting that these nutrients in rainwater were increased after passing through canopies and stems. The percentages of nutrients reaching the forest floor from bulk precipitation were 52%, 43%, 46%, 17%, 35%, and 87% for $Cl^-$, $SO_4^{2-}$, $Ca^{2+}$, $K^+$, $Mg^{2+}$, and $Na^+$, respectively. The bulk precipitation inputs of $NO_3^-$ and $NH_4^+$ during the growing season were 0.53 and 0.31 kg ha$^{-1}$, and the total inputs of these nutrients were and 0.42 and 0.29 kg ha$^{-1}$, respectively. In contrast, the rainfall loading values for $NO_3^-$ and $NH_4^+$ were −0.10 and −0.02 kg ha$^{-1}$, respectively.

**Table 3.** Macronutrients fluxes in water of different type of pathways in comparison to total input, and rainfall loading during the vegetation season.

| Parameter | Nutrient Input (kg ha$^{-1}$) | | | | | | | |
|---|---|---|---|---|---|---|---|---|
| | $Cl^-$ | $NO_3^-$ | $SO_4^{2-}$ | $NH_4^+$ | $Ca^{2+}$ | $K^+$ | $Mg^{2+}$ | $Na^+$ |
| Bulk precipitation | 0.52 | 0.53 | 1.63 | 0.31 | 1.15 | 0.74 | 0.17 | 0.21 |
| Throughfall | 0.99 | 0.42 | 3.75 | 0.29 | 2.43 | 4.23 | 0.48 | 0.23 |
| Stemflow | 0.013 | 0.007 | 0.047 | 0.006 | 0.041 | 0.077 | 0.006 | 0.006 |
| Total input | 1.00 | 0.43 | 3.80 | 0.29 | 2.47 | 4.30 | 0.49 | 0.24 |
| Rainfall loading | 0.48 | −0.10 | 2.17 | −0.02 | 1.33 | 3.56 | 0.32 | 0.03 |

## 4. Discussion

Our results show a very clear enhancement of throughfall nutrient concentrations relative to those in bulk precipitation, as well as an increase in stemflow. However, not all nutrient concentrations were highest in stemflow. These results are in accordance with findings from previous studies [7,37–39].

The chemical composition of throughfall and stemflow is determined by inputs of atmospheric deposition and exchange with vegetation [22,40]. Plant organs serve as temporary storage places for atmospheric deposition. By flowing through the vegetation surface, rainfall washes off particles and gases deposited on the vegetation surface, thus promoting the leaching of nutrients from plant tissues. These two effects are greater than the effects of absorption by plants. Specifically, there is a longer contact time with the vegetation surface during the generation processes of stemflow. On one hand, bark tissue has greater leachability relative to foliage, and stemflow would thus receive more washing-off of dry deposition [38]. On the other hand, the longer residence time on plant organs will favor ion exchanges between rainwater and plant organs [25]. Therefore, stemflow is usually more enriched in nutrients than throughfall.

There was obvious seasonal variation in throughfall and stemflow chemistry at the study site. Most of the nutrient concentrations were very low at the beginning of the growing season, but became more concentrated over time. This is because the younger leaves were able to absorb a large amount of the nutrients dissolved in rainwater [34]. After leaves reached maturity, their ability to absorb nutrients dropped dramatically, and more solutes are left in throughfall and stemflow as a result. Overall, nutrient concentrations were lowest during the middle of the growing season, and showed a dramatic rise towards the end of the growing season. Rainfall events were most severe and most frequent during the middle of the growing season, making atmospheric suspended matter and particulates of plant surface diluted. While at the beginning and end of the growing season, rainfall events were less severe and infrequent, atmospheric suspended matter was greatest, and the time required for accumulation of plant surface particulates was longer. As a result, the EC and solute concentrations were higher.

Due to variation in plant morphology, the initiation and rate of production for stemflow may differ significantly among tree species [41–44]. In addition, there are notable differences in the absorption ability and material composition of the different plant organs and epiphytic organisms [27]. For these reasons, stemflow yield and chemistry can vary significantly depending on the tree species. Nutrient concentrations are closely correlated to tree bark morphology, and the variation in bark morphology can create inconsistencies in the retention rate of rainwater [25,38,45]. Indeed, Parker [46] showed that nutrients appear to be more concentrated in rough-barked species than in smooth-barked species. In our study, nutrients were more concentrated in the stemflow fluxes of deciduous species than in evergreen species. The bark of the two studied evergreen species is smooth with shallow cracks, and the branches are vertical, whereas the two deciduous species have rough bark with deep cracks, and their branches are relatively horizontal. There are also lower frequencies and less stemflow production for rough-barked species compared with smooth-barked species [41,47,48]. Moreover, the relatively acute branching angles of the two evergreen species are conducive to the transfer of rainwater, while the two deciduous species have more horizontal branches that restrict the circulation

of stemflow [30]. Therefore, stemflow fluxes have a larger contact area and longer contact time when flowing through deciduous species. In addition, the barks of deciduous species have more dead tissue. The dead tissue may release large quantities of nutrients (such as $Cl^-$), which would be dissolved in stemflow. All these reasons make stemflow of deciduous tree species have higher washing-off and leaching abilities from plant tissues.

We observed a high degree of variation in nutrient input in the rainfall partitioning of the mixed evergreen and deciduous broadleaved forest. Most nutrients increased in throughfall and stemflow compared to bulk precipitation. $SO_4^{2-}$, $Ca^{2+}$, $Cl^-$, $Mg^{2+}$, and $Na^+$ in bulk precipitation contributed 35 to 87% of the total inputs in the forest floor, while for $K^+$ the value was only 17%. Bulk precipitation is a significant source of nutrient input to the forest, and is especially important for $SO_4^{2-}$, $Ca^{2+}$, $Cl^-$, $Mg^{2+}$, and $Na^+$. The rainfall loadings for $NO_3^-$ and $NH_4^+$ during the growing season were negative values (Figure 3), indicating that the canopies and bark may have strong nitrogen absorption ability. This finding is in line with previous studies that limiting nutrients, such as nitrogen and phosphorus, are usually removed from rainwater as they pass through the vegetation surface [37,49,50]. Besides, the nitrification and denitrification in the vegetation surface may also have impact on nitrogen forms [51]. Our study suggests that understory rainfall tends to accelerate nitrogen utilization during the generation processes.

Although the highest solute concentrations were found in stemflow, the solute fluxes for throughfall were larger, and their contributions to soil fertility are likely more important. In contrast to throughfall, the highly enriched localized inputs of nutrients in stemflow can be directly transferred to the rhizosphere of trees [39,45]. Moreover, the soils at the study site are soft, giving stemflow nutrients the ability to infiltrate rapidly and be absorbed quickly by plants. As a result, stemflow nutrient input enhances nutrient utilization efficiency and cycling rates, and serves as a vital nutrient source for plant growth and development.

## 5. Conclusion

Electrical conductivity and nutrient concentrations were greatest for stemflow, followed by throughfall and bulk precipitation. The ability of stemflow to wash off dry deposition and to stimulate ionic exchange ability results in more concentrated stemflow flux. We found that nutrients were more concentrated in the stemflow fluxes of deciduous species than in evergreen species. This could be ascribed to the rough bark of deciduous species, which stores greater amounts of water, generates less stemflow flux, and extends the residence time of intercepted rainwater. Due to the observed variation in precipitation patterns and vegetation growth, the chemistry of throughfall and stemflow can be viewed as variable. Bulk precipitation contributed significantly to nutrient input, and most nutrients were enriched during passing through the vegetation. In contrast, $NO_3^-$ and $NH_4^+$ were retained along the pathway of rainwater flowing down to the forest floor.

**Author Contributions:** Z.X. and L.S. conceived the idea and designed the experiment. L.S., C.Z. and W.X. performed the experiment. C.Z. and W.X. analyzed data. L.S. and Z.X. wrote the manuscript.

**Funding:** This work was supported by the National Natural Science Foundation of China (41807158), and by the Frontier Science Key Research of the Chinese Academy of Sciences (QYZDY-SSW-SMC011).

**Acknowledgments:** We gratefully acknowledge the staff of Shennongjia Biodiversity Research Station for their assistance in site maintenance and data collection. We would also like to thank Murphy Stephen at the Yale University for his assistance with English language and grammatical editing of the manuscript.

**Conflicts of Interest:** The authors declare no conflict of interest.

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
