# Peer review of "Hydrochemical Fluxes in Bulk Precipitation, Throughfall, and Stemflow in a Mixed Evergreen and Deciduous Broadleaved Forest"

_forests, doi:10.3390/f10060507_

Round 1

Reviewer 1 Report

The article offers a rather small contribution to the current knowledge and its context appears to not have been thoroughly reflected on. The quality differentiation of rainwater in the 'total rainfall' and the water runoff from the trees is obvious. Referring to climate change in the ‘Introduction’ is nice but, to put it crudely, is not enough to publish this article which has a strictly monitoring character. The title is too general. The aims of the article nearly too straightforward and the results obvious. A hypothesis and also scientific novelty are missing.

The manuscript has two major drawbacks:

1. The authors write: “Samples were filtered with filter paper, placed into plastic bottles, and preserved …”. If ‘paper filters’ were used to filtrate, then ‘solute’ (correctly dissolved or soluble) forms of elements/compounds were not analysed. Filtration of water to obtain the soluble forms of components using filter paper is a SERIOUS ERROR. Use of paper filters does not give the possibility to separate suspended solids to the degree required by the method and the popularly accepted norms (see ISO methods or Standard Methods ... APHA).

Here the membrane filters with pore 0.45 µm should have been used. Because of this error, it is not possible to compare the results of this study with others in the existing literature.

2. The pH parameter is lacking – this is a key parameter for understanding the influence of trees on the chemical transformations of the quality of rainfall. The lacking knowledge on this topic is to article’s clear disadvantage and makes its lacking scientific quality so the more clear. It is unlikely that measurements conducted with the YSI 6600V2 did not include pH.

Because the study was located in the transition zone between the northern subtropical and mid-subtropical zone, and because we have similar zones also on other continents, the researchers should pay greatest possible attention to how their study could be used and contribute to research on forest ecosystems worldwide.

Language: The article is written carelessly. Many paragraphs contain non-scientific formulations. The text of the ‘Results’ section is difficult to understand, convoluted, unclear. Most of it is a repetition of the results presented in the graphic form.

Lacking description of symbols (a, b, c in the upper index) in Table 1 and 2 makes it impossible to understand the meaning of the results and the discussion.

Discussion: Many research results are not discussed in the ‘Discussion’. It would be desirable for the authors to discuss the impact of the different contents of chemical elements on the results presented in Tab. 3. The scope of analysis appears to not have been sufficiently thought through, at least in the case of chlorides which are a conservative component of the environment – they are not emitted into the atmosphere, they are not bound by soil and bottom sediments, they do not form permanent compounds with substances dissolved in water, they are only taken by organisms from which, after their death, they are released.

To sum up, the authors would have to revise and expand the article, once again do the analysis of the results and write a new discussion that would also include the anthropopressure or human impact on the pollution of rainfalls.

It would be necessary to have the article checked by a professional native-speaker proofreader specialized in hydrochemistry and natural sciences more generally.

Major revisions must be implemented before the manuscript can be considered for publication. Also the structure of manuscript has to be changed.

For detailed comments please refer to the manuscript.

Author Response

Comments and Suggestions for Authors

The article offers a rather small contribution to the current knowledge and its context appears to not have been thoroughly reflected on. The quality differentiation of rainwater in the 'total rainfall' and the water runoff from the trees is obvious. Referring to climate change in the ‘Introduction’ is nice but, to put it crudely, is not enough to publish this article which has a strictly monitoring character. The title is too general. The aims of the article nearly too straightforward and the results obvious. A hypothesis and also scientific novelty are missing.

Response: Thanks very much for your time on our manuscript and the opportunity to revise our manuscript. We took these comments and suggestions seriously and addressed each of them in every detail. We hope these revisions have resolved the problems and uncertainties pointed out by you. The title has been changed to “Hydrochemical Fluxes in Bulk precipitation, Throughfall, and Stemflow in a Mixed Evergreen and Deciduous Broadleaved Forest”. We have revised the aims of the article and added a hypothesis. In addition, we also revised the results to make the manuscript meet the need of the Journal.

The manuscript has two major drawbacks:

1. The authors write: “Samples were filtered with filter paper, placed into plastic bottles, and preserved …”. If ‘paper filters’ were used to filtrate, then ‘solute’ (correctly dissolved or soluble) forms of elements/compounds were not analysed. Filtration of water to obtain the soluble forms of components using filter paper is a SERIOUS ERROR. Use of paper filters does not give the possibility to separate suspended solids to the degree required by the method and the popularly accepted norms (see ISO methods or Standard Methods ... APHA).

Here the membrane filters with pore 0.45 µm should have been used. Because of this error, it is not possible to compare the results of this study with others in the existing literature.

Response: The filters used in our study are showed in the following photos. I don’t know filters very well, so miscall them filter paper. We have revised the manuscript and added the type of the filter.

2. The pH parameter is lacking – this is a key parameter for understanding the influence of trees on the chemical transformations of the quality of rainfall. The lacking knowledge on this topic is to article’s clear disadvantage and makes its lacking scientific quality so the more clear. It is unlikely that measurements conducted with the YSI 6600V2 did not include pH.

Response: The pH values were monitored with YSI 6600V2. One reviewer suggested us to omit pH values since there was no significant difference in different rainfall fractions. We have added pH values in the revised manuscript.

Because the study was located in the transition zone between the northern subtropical and mid-subtropical zone, and because we have similar zones also on other continents, the researchers should pay greatest possible attention to how their study could be used and contribute to research on forest ecosystems worldwide.

Response: The work is focus on the special forest – mixed evergreen and deciduous broadleaved forest. We have read the papers in similar zones. We are writing a review paper about the rainfall partitioning and related nutrient fluxes in subtropical areas, and we will expand the discussion in that paper. Besides, we have preliminarily compared the rainfall fraction of the forest with the forests in other subtropical areas and other forest in our previous papers.

Language: The article is written carelessly. Many paragraphs contain non-scientific formulations. The text of the ‘Results’ section is difficult to understand, convoluted, unclear. Most of it is a repetition of the results presented in the graphic form.

Response: We have revised the manuscript to make the manuscript more readable and use scientific formulations as possible as we can.

You think it is not suitable to have a repetition of the results presented in the graphic form. In our general understanding, the results section should highlight or supplement the results presented in the graphic form. So we are puzzled about it, we revised the results section and tried to not simply repeat the information that could be easily got in the graph.

Lacking description of symbols (a, b, c in the upper index) in Table 1 and 2 makes it impossible to understand the meaning of the results and the discussion.

Response: The letters could represent the difference in values. Values with same letters suggest that there is no significant difference among these values, Values with different letters suggest that there is significant difference among these values. This is a generally accepted usage in statistical analysis, so we think it is not necessary to describe these symbols.

Discussion: Many research results are not discussed in the ‘Discussion’. It would be desirable for the authors to discuss the impact of the different contents of chemical elements on the results presented in Tab. 3. The scope of analysis appears to not have been sufficiently thought through, at least in the case of chlorides which are a conservative component of the environment – they are not emitted into the atmosphere, they are not bound by soil and bottom sediments, they do not form permanent compounds with substances dissolved in water, they are only taken by organisms from which, after their death, they are released.

To sum up, the authors would have to revise and expand the article, once again do the analysis of the results and write a new discussion that would also include the anthropopressure or human impact on the pollution of rainfalls.

Response: Thanks for your valuable suggestion to improve our manuscript, we have make effect to discuss all useful results to broaden the discussion section. Your comments on the emission of nutrients of dead organisms help us explain why there is difference in stemflow chemistry between evergreen and deciduous tree species.

It would be necessary to have the article checked by a professional native-speaker proofreader specialized in hydrochemistry and natural sciences more generally.

Response: We have revised the whole manuscript carefully and tried to avoid possible grammar error. Besides, we have asked two native English speakers to check the English.

How does DBH (diameter at breast height) make sense, if in the results you given't data on the water quality differentiation according to DBH.

Response: We chose trees at different DBH classes to determine stemflow yield, not to compare the stemflow chemistry among different tree DBH classes. It is a very interesting idea to analyze the variation in stemflow chemistry among different tree DBH classes (tree age), so we decide to do this study and design more reasonable experiment in future work.

Major revisions must be implemented before the manuscript can be considered for publication. Also the structure of manuscript has to be changed.

Response: We have revised this manuscript according to your suggestion in the PDF file. Your suggestion is constructive for improving our manuscript. The revised details are showed in the revised manuscript. Most of your comments are very helpful for our manuscript, but we are a little confused about a few comments.

Average annual precipitation = unit should be mm/m2. Change in whole paper!

Response: Thanks very much for pointing out this problem. However, in our opinion, annual precipitation is the depth of rainwater that one area could receive in one year, so the unit of annual precipitation should be the unit of depth. After receiving your comment, we read many papers and materials to identify the unit of precipitation, all of the authors take mm as the unit of annual precipitation.

Are these local studies? If not, add a map of whole Asia instead of the province map.

Response: The study focus on a special forest. Therefore, strictly speaking, it is not a local study. We have presented China in central Asia, and we think the map is enough to show the location of the study area. We draw the province map just to present the detailed address of the study site.

Imprecise description of figure 2 see here percentage share.

Response: The data in Figure 2 (Figure 3 in the revised manuscript) is indeed input (the unit is mm), not percentage (%). We presented monthly input of throughfall, stemflow, and interception loss, and described the percentage and input of them in text.

Reviewer 2 Report

This paper is well presented and gives a contribution to the understanding  of chemistry  of  the   hydrological fluxes in mixed evergreen forests and broad leave forests., in particular,  stemflow  diferences between the two forests types.

This  paper investigate the chemistry of gross rainfall, stemflow and throughfall in a mixed evergreen and deciduous broadleaved forest in  Central China. Mean solute concentrations in throughfall and stemflow were higher than in gross rainfall. Stemflow ion fluxes from deciduous tree species were greater than those for evergreen trees . Throughfall and stemflow chemistry fluctuated dramatically over the growing season. Inorganic nitrogen (NO3- and NH4+) was retained, while other solutes were washed off or leached via throughfall and stemflow pathways.

Comments

Contribution for the understanding of chemistry of  hydrological fluxes in  mixed evergreen forests and deciduous forests.

This study is original since it adresses the differences in stemflow chemistry between evergreen and deciduous broadleaved tree species.The approach is well defined.

The results  are interpreted adequately and are  significant •

The data and analyses are well presented •

The experimental design is adequated  based in a single growing season. Even so, it provided enough data  for the analyses.

Nevertheless it is not clear if this   season (May –October) 2014 is well representative of a “normal” rainfall year. Nothing is said in the paper regarding  rainfall  inter annual variability. 

Author Response

Comments and Suggestions for Authors

This paper is well presented and gives a contribution to the understanding of chemistry of  the hydrological fluxes in mixed evergreen forests and broad leave forests., in particular, stemflow diferences between the two forests types.

This paper investigate the chemistry of gross rainfall, stemflow and throughfall in a mixed evergreen and deciduous broadleaved forest in Central China. Mean solute concentrations in throughfall and stemflow were higher than in gross rainfall. Stemflow ion fluxes from deciduous tree species were greater than those for evergreen trees. Throughfall and stemflow chemistry fluctuated dramatically over the growing season. Inorganic nitrogen (NO3- and NH4+) was retained, while other solutes were washed off or leached via throughfall and stemflow pathways.

Comments

Contribution for the understanding of chemistry of hydrological fluxes in mixed evergreen forests and deciduous forests.

This study is original since it adresses the differences in stemflow chemistry between evergreen and deciduous broadleaved tree species. The approach is well defined.

The results are interpreted adequately and are significant.

The data and analyses are well presented.

The experimental design is adequated based in a single growing season. Even so, it provided enough data for the analyses.

Nevertheless it is not clear if this season (May –October) 2014 is well representative of a “normal” rainfall year. Nothing is said in the paper regarding rainfall inter annual variability.

Response: Thank you for your recognition and approval of our work. The season (May–October) in 2014 is a relative rainy season. The rainfall input during this period is 1374 mm, while the average annual precipitation is 1350 mm, and the growing-season (May–October) precipitation is 1148 mm. The precipitation of this growing season is 20% higher than the mean value. Although it is not a very well representative of normal rainfall year, we still could take the result as a fair description of the forest.

Round 2

Reviewer 1 Report

Answers and supplements are sufficient. Quality of manuscript now is significantly higher than earlier. Currently, the work contains only minor shortcomings that require changes (see pdf file). Corrections will improve the readability and clarity of the scientific message.

Author Response

Comments and Suggestions for Authors

Answers and supplements are sufficient. Quality of manuscript now is significantly higher than earlier. Currently, the work contains only minor shortcomings that require changes (see pdf file). Corrections will improve the readability and clarity of the scientific message.

Response: Thanks very much for your professional opinions and pertinent evaluations. We took all your comments seriously and they greatly contributed to improving our manuscript.

Letters indicate level of P-value: a <0.05, b >0.05, c ???.

Response: It is a widely accepted expression in statistical analysis. Values with the same letters suggest that there are no significant difference among these values. If values are followed by different letters, there are significant difference among these values.

Please see examples in the following articles:

Dawoe, E.K.; Barnes, V.R.; Oppong, S.K. Spatio-temporal dynamics of gross rainfall partitioning and nutrient fluxes in shaded-cocoa (Theobroma cocoa) systems in a tropical semi-deciduous forest. Agroforest. Syst. 2018, 92, 397–413.

Silva, I.C.; Rodríguez, H.G. Interception loss, throughfall and stemflow chemistry in pine and oak forests in northeastern Mexico. Tree Physiol. 2001, 21, 1009–1013.

Germer, S.; Zimmermann, A.; Neill, C.; Krusche, A.V.; Elsenbeer, H. Disproportionate single-species       contribution to canopy-soil nutrient flux in an Amazonian rainforest. For. Ecol. Manag. 2012, 267, 40–49.
